# Groin Surgical Site Infection in Vascular Surgery: Systemic Review on Peri-Operative Antibiotic Prophylaxis

**DOI:** 10.3390/antibiotics11020134

**Published:** 2022-01-20

**Authors:** Bruno Amato, Rita Compagna, Salvatore De Vivo, Aldo Rocca, Francesca Carbone, Maurizio Gentile, Roberto Cirocchi, Francesco Squizzato, Andrea Spertino, Piero Battocchio

**Affiliations:** 1Department of Public Health, University Federico II of Naples, 80131 Naples, Italy; francesca.carbone@unina.it; 2Division of Vascular Surgery, Ospedale Pellegrini, 80100 Naples, Italy; rita.compagna@libero.it (R.C.); sdevivo71@gmail.com (S.D.V.); 3Deparment of Medicine and Health Sciences “V. Tiberio”, University of Campobasso, 86100 Campobasso, Italy; aldo.rocca@unimol.it; 4Department of Clinical Medicine and Surgery, University Federico II of Naples, 80131 Naples, Italy; magentile@unina.it; 5Department of General and Oncologic Surgery, University of Perugia, 05100 Terni, Italy; roberto.cirocchi@unipg.it; 6Department of Cardiac, Thoracic and Vascular Sciences, University of Padova, 35100 Padova, Italy; francesco_squizzato@icluod.com (F.S.); andrea.spertino@gmail.com (A.S.); piero.battocchio@yahoo.it (P.B.)

**Keywords:** systemic review, vascular surgery, groin infection, antibiotic therapy

## Abstract

Objectives: Surgical site infections (SSIs) in lower extremity vascular surgeries, post-groin incision, are not only common complications and significant contributors to patient mortality and morbidity, but also major financial burdens on healthcare systems and patients. In spite of recent advances in pre- and post-operative care, SSI rates in the vascular surgery field remain significant. However, compliant antibiotic therapy can successfully reduce the SSI incidence pre- and post-surgery. Methods: In October 2021, we conducted a systematic literature review using OVID, PubMed, and EMBASE databases, centered on studies published between January 1980 and December 2020. The review adhered to the Preferred Reporting Items for Systematic Reviews and Meta Analyses checklist. Inclusion/exclusion criteria have been carefully selected and reported in the text. For analyses, we calculated 95% confidence intervals (CI) and weighted odds ratios to amalgamate control and study groups in publications. We applied The Cochrane Collaboration tool to assess bias risk in selected studies. Results: In total, 592 articles were identified. After the removal of duplicates and excluded studies, 36 full-texts were included for review. Conclusions: The review confirmed that antibiotic therapy, administered according to all peri-operative protocols described, is useful in reducing groin SSI rate in vascular surgery.

## 1. Introduction

Surgical site infections (SSIs) are major concerns for all surgical specialties, with the literature reporting SSI risks of 2–4% for “clean surgery” [1]. Higher rates have been reported for post-traumatic procedures (15–50%), or in selected populations, including high-risk vascular surgery patients (15–22%), and accompanied by a considerable lengthening of hospitalization times, a high mortality rate (26–67%), and the cost-burden to community [2,3]. Similarly, SSI-mediated morbidity may be higher when prosthetic grafts are used in complex surgeries; thus, SSI prevention and treatment are both clinically significant. The current recommendations for clean vascular surgical procedures advocate no more than 24 h of intravenous antibiotic post-operative therapy, as no benefits are indicated past this treatment period [4,5,6]. However, when overt infection signs or risks are present, in particular, when synthetic prostheses are concerned, the literature is less clear on this topic. Therefore, as these indications require clarification, interest continues to be strong in the use of new antibiotics and also in alternative methods of their delivery.

In vascular surgery, lower extremity bypass surgery for limb salvage has the highest rate of SSI incidence, with rates varying between 5% and 30% [1,2,3,4,5,6]. SSIs increase hospital stays, increase readmission rates, incur elevated mortality and morbidity rates, increase healthcare burdens, and increase the incidence of repeat revascularization surgeries [4,5]. Moreover, several health risk factors such as diabetes, smoking, being female, prosthetic grafts, obesity, and steroid use contribute to lower extremity SSIs [7,8,9,10,11]. 

While some risk factors are not modifiable, identifying modifiable factors can successfully reduce SSIs. In particular, the rapid identification of causative bacteria is vital to establish and select the most appropriate antibiotic therapy. While any bacteria can theoretically infect a vascular prosthesis or contribute to an SSI, the Gram-negative *Pseudomonas aeruginosa* and Gram-positive *Staphylococcus epidermidis* and *Staphylococcus aureus* strains are the most commonly found in SSIs of the groin, while fungal infections are found less frequently, although present in the inguinal area [8,12,13,14,15]. A considerable challenge to SSI diagnostics and treatment are biofilms (or microfouling) [16,17]: these complex bacterial polymeric aggregations, consisting of bacterial glycocalyx with incorporated microcolonies, are characterized by the secretion of protective matrix adhesives that provide safe environments from external antimicrobial agents and host defenses [16]. Biofilms become manifest as polymicrobial infections and contain dominant, highly resistant, difficult-to-eradicate bacteria, which increase patient mortality and morbidity [18].

As in all surgical specialties, vascular SSI bacterial typology has changed over time, and is reflected by an elevated incidence of antibiotic-resistant bacteria, especially Staphylococcal family members. In particular, methicillin-resistant *S. aureus* (MRSA) infections are a serious risk factor for nosocomial-morbidity and morbidity in terms of admission to intensive care, repeated surgical procedures, and major amputation infection risks [19,20,21,22]. Rarely, isolated strains of *Staphylococcus aureus* are still sensitive to penicillin. More frequently, these Staphylococci are distinguished as MRSA (methicillin-resistant *Staphylococcus aureus*) and MSSA (methicillin-sensitive *Staphylococcus aureus*). Resistance to penicillin (MSSA) is conferred by a bacterial penicillinase. This resistance can be overcome by adding a beta-lactamase inhibitor (e.g., amoxicillin/clavulanic acid, ampicillin/sulbactam) or by using a penicillinase-resistant penicillin (e.g., oxacillin). Methicillin resistance (MRSA) is conferred by the presence of the bacterial gene mecA, which codes for a penicillin-binding protein, an enzyme that has a low affinity for beta-lactams, and therefore leads to resistance to methicillin and oxacillin. There are MRSA of hospital origin often characterized by extended resistance to antibiotics (MDR, multidrug resistance) and MRSA that is community-acquired, CA-MRSA; the latter of which can maintain sensitivity to tetracyclines (tetracycline, doxycycline, minocycline, tigecycline) [23]. 

In recent years, the WHO Report on Surveillance of Antibiotic Consumption (2016–2018) described how the percentage of MRSA, in the evaluation of hospital infections, has remained stable (around 34%). Relative to Gram-positive bacteria, the highest resistance rates were observed for *S. aureus* to erythromycin (38.9%), clindamycin (34.4%), methicillin (33.5%), and levofloxacin (31.5%). For many years, the treatment of choice to combat MRSA has been based on the use of glycopeptides (vancomycin and teicoplanin); however, the excessive and careless use of these antibiotics has led to the emergence of strains with decreased sensitivity to vancomycin. In recent years, new antibiotics have been introduced into clinical practice, such as linezolid, daptomycin, and more recently, ceftaroline, also in combination with vancomycin and daptomycin, for the treatment of severe MRSA infections. For the latter antibiotics too, particularly linezolid and daptomycin, the emergence of resistant strains has been observed [24]. Thus, active SSI prevention is highly advisable for peri-operative antibiotic therapy, in addition to the usual stringent antiseptic and sterility policies. Such combinations should minimize vascular SSI onset, especially in inguinal areas.

## 2. Materials and Methods

Two independent authors (CR and CF) performed a systematic literature search using OVID, PubMed, and EMBASE databases in October 2021. The search centered on studies written in English and published between January 1980 and December 2020. Our strategy adhered to the Preferred Reporting Items for Systematic Reviews and Meta Analyses guidelines. 

Search terms: “surgical site infection” AND “vascular surgery” AND “antibiotic therapy”. Bibliographies of selected studies were examined to identify other potentially relevant articles. Inclusion criteria were documents, books, clinical trials, randomized controlled trials (RCTs), systematic reviews, reviews, observational studies, and meta-analyses.

Two co-authors (AR and BA) independently conducted screening, review, and quality assessments. When eligibility disagreements arose, another author (MG) reviewed the study, and agreement was reached. We calculated 95% confidence intervals (CI) and weighted odds ratios to amalgamate and analyze control and study groups in studies. We combined treatment effects using the Chi-square test, and Mantel–Haenszel risk ratios were used to assess heterogeneity. To assess the risk of bias for selected studies, the Cochrane Collaboration tool was used.

## 3. Results

We identified 592 publications. After the removal of 228 duplicates, 364 publications were examined for inclusion and exclusion criteria (Figure 1). The screening strategy is also shown in Figure 1. 

## 4. Study Descriptions

We selected 36 full-text articles for inclusion: 2 meta-analyses, 2 systematic reviews, 11 clinical trials, 7 reviews, and 14 observational studies (mostly retrospective) were analyzed (Figure 1). Articles were deemed highly relevant if they specifically dealt with commonly used preventative and treatment antibiotic therapies for SSIs after groin incision in vascular surgery.

## 5. Risk of Bias

In 13 studies, authors satisfactorily outlined randomization and allocation masking methods. Therefore, these studies were deemed at low risk of selection bias (Table 1).

We deemed 32 studies at high risk of bias as outcome assessments were not blinded. We considered 25 studies at low risk of selective reporting, but for the other 21 studies, this was unclear. 

## 6. Pre-Operative Antibiotic Therapy

Although SSIs are relatively rare in vascular surgery, they have serious effects. Thus, pre-operative infection prophylaxes are advised to minimize infection risks, especially for patients with synthetic grafts. To address this, we analyzed the evidence on pre-operative antibiotics prior to vascular surgery. Firstly, for most vascular interventions, Gram-positive bacteria, in particular *S. aureus*, represent approximately 80% of all SSIs [2,3]. In contrast, Gram-negative bacteria are implicated in 20% to 25% of infections [1,2,3,4]; however, for effective antibiotic prophylaxis, both bacterial types should be targeted and controlled.

Synthetic vascular prosthesis implantation generates particular microenvironments and conditions that can promote bacterial wound invasion and biofilm formation; this latter functionality defends enclosed bacteria from antibiotic therapy and host defense [38]. To reduce or eliminate the bacterial colonization of damaged tissues, an antibiotic prophylaxis should be administered for effective drug concentrations in tissue at the surgical site before surgery commences. Therefore, the administration of pre-operative antibiotic therapy is advisable 60 min before surgery, and also in additional intraoperative doses if the operation lasts more than 4 h and/or more than 1500 cc blood is lost [25,38].

In their 34 RCT meta-analysis, Stewart et al. investigated the effects of systemic antibiotics compared with placebo in patients requiring synthetic graft vascular surgery, with overall consistent SSI reductions identified using systemic antibiotics (relative risk = 0.24; 95% CI: 0.16–0.38; *p* < 0.001). These authors observed clear benefits of prophylactic antibiotic use for arterial surgery reconstruction, while other interventional approaches (pre-operative bathing with antiseptic agents and wound drainage) lacked effective evidence [39].

Moreover, a prospective, randomized, blinded study by Pitt et al. showed that either pre-operative systemic or intraoperative topical antibiotics were largely effective in preventing groin wound SSIs, but no therapeutic advantages were observed when treatments were combined [40].

To combat SSIs in vascular surgery, the prophylaxes of choice are primarily first or second-generation cephalosporins, as they are low-cost options (the most common prophylaxis is cefazolin). Vancomycin or clindamycin appear to be preferentially used for patients with β-lactam allergy [38]. 

It is useful to remember that allergies related to common antibiotics such as cephalosporins are among the most frequently reported allergies, but only 10% of these patients are truly allergic to these drugs. This situation leads to potential unintended damage for patients and creates difficulties for doctors’ treatment decisions. The detection of actual allergies with skin tests is therefore important for decision-making strategies in these patients [25].

Gram-negative bacterial coverage is also an important factor as groin and abdominal surgeries are often infected with gastrointestinal tract flora [41]. While other studies [25,26] compared single- and multiple-dose antimicrobials, no significant differences were identified.

Due to the higher prevalence of MRSA, infection trends are changing [15]. In hospitals with high MRSA prevalence rates, or patients at high infection risk (e.g., geriatric, oncological, and dialysis patients), vancomycin may be used for prophylaxis. When this antibiotic is administered with cefazolin, it is more effective in stopping SSI and targeting Gram-negative bacteria [41]. 

Stone et al. compared cefazolin with the association of cefazolin plus daptomycin administration to prevent pre-operative vascular SSIs and identified lower SSI rates in the combined antibiotics cohort (3.9% versus 12.9%) [28]. In a similar study, these authors compared combined cefazolin/daptomycin with cefazolin/vancomycin in patients undergoing vascular surgery. The combined vancomycin group showed decreased Gram-positive SSI rates during the post-operative course [27].

Mohammad et al. demonstrated that locally applied intraoperative vancomycin plus standard pre-operative antibiotics lowered groin wound SSIs in patients undergoing arterial operations. Although this positive result mainly reflected a reduction in superficial SSIs, the deep wound dehiscence rate did not show a significant decrease [42]. Thus, from these studies, anti-MRSA prophylaxes should be selectively administered to high-risk MRSA infection populations.

Classen et al. showed that cephalosporin supplement administration was favorable for conditions where blood loss exceeded 1500 cc. and/or surgery lasted more than 240 min [29]. In the event of a cephalosporin-related allergy, vancomycin or clindamycin should be considered as substitutes, along with adequate Gram-negative coverage. Aztreonam is considered an appropriate substitute [41].

For the prophylactic administration of vancomycin, the drug should be administered 60–120 min prior to surgery as drug tissue distribution rates and bactericidal activities are slower than cephalosporins [1]. Pounds et al., in their retrospective study on hospitalization stays before surgical procedures, showed that SSI rates were higher for prolonged pre-operative hospitalizations. Therefore, avoidable hospitalization before surgery should be ensured [15].

For endovascular revascularization procedures, there is no general consensus on prophylaxis with antibiotics. Infection rates after endovascular aortic repair or thoracic endovascular aortic repair procedures occur in less than 1% of patients [30,43], whereas infection rates are higher in emergency settings [28]. However, endovascular graft infections are related to high mortality and morbidity [44]. Antibiotic prophylaxis corresponding to that performed for open vascular procedures, through a single pre-operative dose of cefazolin and vancomycin (as an alternative for patients with β-lactamase allergy) is therefore recommended and widely practiced [44] Deikema et al. [45] reported that routine MRSA nasal carriage screening putatively reduced MRSA infection risks in patients having major surgery; however, the data were not strong. In a large-scale, retrospective interventional cohort study, Harbarth et al. [31] showed that for patients undergoing routine MRSA carriage screening before surgery (patients received one 5-day trial of mupirocin nasal ointment and a pre-operative prophylactic antibiotic for MRSA), no differences in MRSA SSI rates were observed when compared with controls. In contrast, Malde et al. [46] supported routine MRSA screening in those undergoing vascular surgery: this retrospective study demonstrated significant reductions in MRSA infections and amputation rates after routine MRSA screening was implemented. 

This evidence suggests that adequate pre-operative antibiotic administration, with reference to bactericidal and pharmacokinetic activities and associated Gram-positive and -negative bacteria coverage, may limit groin SSI rates during vascular procedures. Antibiotics should be adapted to the patient’s past infection history and a concerted effort made to avoid a prolonged non-essential pre-operative hospitalization. Based on the evidence from this review, however, routine nasal MRSA screening does not appear to be essential.

Synthetic data are shown in Table 2.

## 7. Intraoperative Use of Local Antibiotics

Optimized aseptic surgery techniques and correct antibiotic prophylaxis are ideal preventative measures against SSIs. Prolonged antibiotic administration may increase resistance risks, allergy, and/or toxicity rates. In addition, tissue trauma during surgery and healing processes may partially reduce tissue perfusion and cause decreased tissue penetration of systemically administered antibiotics [47,48]. Collagen implants containing gentamicin can be used to reduce groin wound SSIs and the requirement for repeated surgery by providing high, local concentrations of antibiotics (gentamicin), but at low levels in serum [32,33,34]. Gentamicin-containing collagen implants limit wound complications and repeated surgeries and are completely absorbable, unlike polymethylmethacrylate (PMMA) spheres, which need to be removed [48]. Furthermore, when gentamicin is used intraoperatively (local), its concentration levels rapidly decrease, thereby avoiding antibiotic resistance [47,48]. Similarly, collagen is advantageous as it simultaneously functions as an adjuvant for hemostasis and for the healing processes [34].

Investigations on collagen implants containing antibiotics (gentamicin) for local intraoperative use have shown they reduce SSIs in patients having undergone general surgery, orthopedic, gynecological, and other general surgeries [35,49]. A single, prospective randomized investigation by Costa Almeida et al. evaluated gentamicin collagen implants for vascular surgery [36]. The study included 60 non-diabetic and non-obese patients treated with gentamicin collagen implants at the groin wound site at the end of vascular prosthetic surgery, compared to a control group. No (0%) SSI was identified in the study group compared with 6 (10%) SSI cases in controls. Moreover, significantly decreased hospital stays were observed between groups. Other smaller studies have produced similar results [37,50], but larger multi-center RCTs are required to validate these observations.

Synthetic data are shown in Table 3.

## 8. Study Limitations

Some limitations can be identified in our study by differences in inclusion/exclusion criteria between the various studies selected for the systematic review. However, heterogeneity indices for each study were negligible. Equally, a risk of selection bias of articles may have occurred. However, the authors stringently observed systematic literature search guidelines to eliminate potential bias. A further study limitation was the blinding of those who produced the evaluation of the wounds: this was, in fact, performed through a subjective assessment of the authors, as double-blind treatment would not have been possible. However, we took steps to minimize this potential bias, at least for RCT.

## 9. Conclusions

SSI risks are major complications associated with inguinal incisions in the vascular surgery field and are also related to significant mortality and morbidity rates. We performed an evidence-based literature review to evaluate and confirm the validity of peri-operative administration of antibiotics for the prevention of SSI in vascular surgery, frequently the source of prosthetic infections and late failure of revascularization procedures. The administration of pre-operative antibiotic prophylaxis (with coverage of Gram-positive and Gram-negative bacteria) and the possible supplement of a second similar dose in the intraoperative phase, if the interventions last more than 4 h and/or the blood loss exceeds 1500 cc, in fact, was found to have a significant impact in reducing inguinal SSI rates during vascular surgery.

Strategically designed RCTs and retrospective investigations should be conducted to corroborate these observations. However, based on this review, we aim to use these observations as standard pre-operative strategies for vascular surgeons in today’s practice. 

In the future, these modifications to our policies will allow us to retrospectively evaluate the effects of these interventions on inguinal SSI rates after vascular interventions, thereby addressing knowledge gaps in the literature.

## Figures and Tables

**Figure 1 antibiotics-11-00134-f001:**
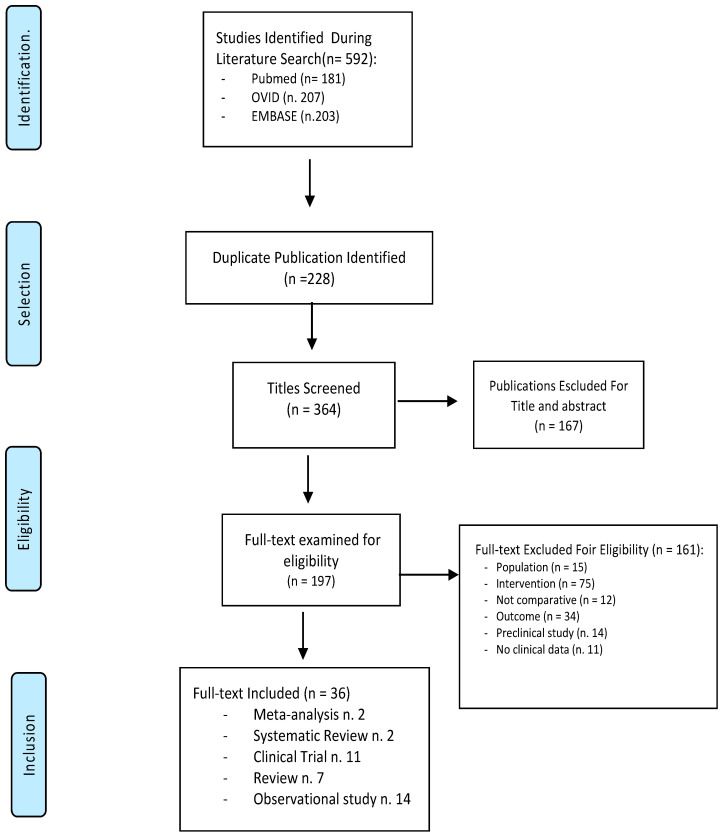
Flowchart for references selection on “surgical site infection” AND “vascular surgery” AND “antibiotic therapy”. STRATEGY: Query: “surgical site infection” AND “vascular surgery” AND “antibiotic therapy”; Results by years: *1980–2020*; Text availability: *Full text*; Article attribute: *Meta-analysis, RCT, Review, Systematic Review, Clinical Trial, Observational Studies, Books and Documents*.

**Table 1 antibiotics-11-00134-t001:** Risk of biass assessment in the included randomized studies.

Study	Random Sequence Generation (Selction Bias)	Allocation Concealment (Selection Bias)	Blinding (Performance Bias and Detection Bias)	Selective Reporting (Reporting Bias)	Other Bias	Jadad Score
Bratzler DW et al., 2005 [25]	High	High	High	Low	Low	2
McDonald M et al., 1998 [26]	Low	Low	High	Low	Low	3
Patrick S et al., 2010 [27]	Low	Low	High	Low	Low	3
Stone PA et al., 2015 [28]	High	High	High	Low	Low	2
Classen DC et al., 1992 [29]	Low	Low	High	Low	Low	3
Argyriou C et al., 2017 [30]	High	High	High	Low	Low	2
Harbarth S et al., 2008 [31]	High	High	High	Low	Low	2
Friberg O et al., 2005 [32]	Low	Low	High	Low	Low	3
Eklund AM et al., 2007 [33]	Low	Low	High	Low	Low	3
Raja SG et al., 2012 [34]	High	High	High	Low	Low	2
Chang WK et al., 2013 [35]	High	High	High	Low	Low	2
Costa Almeida CEP et al., 2014 [36]	High	Low	High	Low	Low	3
Holdsworth et al., 1999 [37]	Low	High	Low	Low	Low	3

**Table 2 antibiotics-11-00134-t002:** Summary of the included study for pre-operative antibiotic therapy.

References	Methods	Partecipants	Intervention	Outcomes	Primary Findings
Johnson JH et al., 1992 [38]	Single blinded, randomized controlled trial	2847 patients undergoing elective clean or “clean-contaminated” surgical procedures	Administration of antibiotics pre- v/s peri- and post-operatively	Surgical-wound infections	SSI rate of 0.6% for pre- vs. 1.4% peri- and 3.3% post-operative antibiotic administration (*p* less than 0.0001; relative risk, 5.8; 95 percent confidence interval, 2.6 to 12.3)
Bratzler DW et al., 2005 [25]	Systematic review and meta-analysis	22 trials of prophylactic systemic antibiotics	Prophylactic systemic antibiotics administration	Wound infection or early graft infection	Prophylactic systemic antibiotics reduced the risk of wound infection (RR, 0.25; 95% confidence interval [CI], 0.17 to 0.38)
Stewart AH et al., 2007 [39]	Guidelines review	Published North American guidelines for antimicrobial prophylaxis until 2002	Pre- and post-operative antimicrobial prophylaxis	Surgical-wound infections	Infusion of the first antimicrobial dose should begin within 60 min before surgical incision and that prophylactic antimicrobial agents should be discontinued within 24 h of the end of surgery
Pitt HA et al., 1980 [40]	Single blinded, randomized controlled trial	217 patients scheduled for vascular surgery with groin incision	No antibiotic v/s topical cephradine prior to closure v/s 24-h perioperative e.v. cephradine and v/s both topical and intravenous cephradine.	Groin and abdominal incisional infections	- Groin and abdominal incisional infections significantly reduced (*p* < 0.01) among patients who received prophylactic antibiotics by either the topical, systemic, or combined routes of administration. - No significant differences were noted among the three antibiotic groups.
Bratzler DW et al., 2013 [41]	Practice guidelines	Primary literature of Therapeutic Guidelines on Antimicrobial Prophylaxis in Surgery	- Single pre-incision dose of cefazolin or cefuroxime- Continuing prophylaxis.	Primary prophylaxis and eradication of wound infection	Recommendation of a single pre-incision dose of cefazolin or cefuroxime with appropriate intraoperative redosing. No evidence for continuing prophylaxis until all drains and catheters are removed. Clindamycin or vancomycin as alternative in patients with b-lactam allergy. Vancomycin used for prophylaxis in patients known to be colonized with MRSA. (Strength of evidence for prophylaxis = A)
McDonald M et al., 1998 [26]	Systematic review	- 28 Clinical Trials - 9478 patients	Antimicrobial single v/s multiple dose in surgical prophylaxis	Post-operative surgical site infections rate prevention	No clear advantage of either single or multiple-dose regimens of antibiotics
Stone PA et al., 2015 [28]	Single center prospective double blinded randomized study	178 patients were evaluated at 90 days for surgical site infection	Vancomycin v/s Vancomycin + Daptomycin pre-operative administration	Post-operative SSI rate prophylaxis	Vancomycin supplemental prophylaxis seems to reduce the incidence of Gram-positive infection compared with adding supplemental Daptomycin prophylaxis (*p* = 0.11).
Patrick S et al., 2010 [27]	Single institution prospective randomized study	169 low-risk patients undergoing elective vascular procedures	Cefazolin, cefazolin + vancomycin, or cefazolin + daptomycin surgical prophylaxis	Post-operative surgical site infections rate prevention	Significant fewer infectious complications in the cefazolin + daptomycin group
Mohammed S et al., 2013 [42]	Retrospective Cohort study	454 patients who underwent open vascular procedures	Systemic vancomycin v/s systemic + local application of vancomycin powder	Inguinal wound infection and dehiscence over a 30-day period	Addition of intraoperative local vancomycin did not improve the rates of inguinal wound dehiscence or deep infections but had a positive impact on superficial wound infections
Classen DC et al., 1992 [29]	Single institution prospective randomized study	2847 patients undergoing elective clean surgical procedures	Antibiotic administration 2 to 24 h before the surgical incision	Post-operative surgical wound infection rate	Prophylactic administration of antibiotics in the two hours before surgery reduces the risk of wound infection.
Cernohorsky P et al., 2011 [43]	Multicenter retrospective cohort study	1431 endovascular procedures	Prophylactic antimicrobial therapy	Incidence of endograft infection and mortality rate	Endograft infection rate below 1%, with a mortality rate of 25%. Antimicrobial therapy helps conservative treatment in selected cases of patients with an infected endograft.
Argyriou C et al., 2017 [30]	Meta-Analysis	12 studies reporting on 362 patients	Endovascular aneurysm repair (EVAR)	Evidence on the outcomes of graft infection after EVAR	Supportive medical antimicrobial treatment without surgical intervention has a significant associated mortality.
Chehab MA et al., 2018 [44]	Practice guidelines	Primary literature of Therapeutic Guidelines on Antimicrobial Prophylaxis Vascular and IR Procedures	Prophylactic antimicrobial therapy	SSI antimicrobial prophylaxis	Recommendation 1: intravenous (IV) antibiotic agents must be administered within 1 h of an incision. Recommendation 2: A repeat dose of antibiotic agents should be administered if a period of 2 h has lapsed from the initial dose.
Diekema DJ. et al., 2007 [45]	Review	Evidence-based guidelines	Program for detection of methicillin-resistant *Staphylococcus aureus* and vancomycin-resistant enterococci among hospital patients	Steps that should be performed when planning active surveillance cultures for detection of methicillin-resistant *Staphylococcus aureus* and vancomycin-resistant enterococci	Preparing the laboratory and reducing the turnaround time for screening tests; monitoring and optimizing the intervention of instituting contact precautions; monitoring and ameliorating the known adverse effects of contact precautions.
Harbarth S et. al., 2008 [31]	Prospective, interventional cohort study. Clinical trial	754 patients	Compare rapid MRSA screening on admission plus standard infection control measures vs. standard infection control alone	Perioperative antibiotic prophylaxis of MRSA carriers and topical decolonization for 5 days.	A universal, rapid MRSA admission screening strategy did not reduce nosocomial MRSA infection in a surgical department with endemic MRSA prevalence but relatively low rates of MRSA infection
Malde DJ et al., 2006 [46]	Retrospective Cohort study	280 vascular patients	Data analysis of two period groups of MRSA positive vascular patients	Wound infection, major limb amputation and mortality rates	MRSA screening identifies patients at risk of serious complications and is associated with a reduction in these complications following both elective and emergency surgery

**Table 3 antibiotics-11-00134-t003:** Summary of the included study for intra-operative collagen-containing gentamicin implantation (CCGI).

References	Methods	Partecipants	Intervention	Outcomes	Primary Findings
Hussain ST et al., 2012 [47]	Review	Five publications on development of SSI in vascular surgery	Prophylactic use of GCCI in fem-pop graft surgery	Reduction in SSI rate incidence	GCCI have a role to play in preventing and treating SSI following vascular reconstruction.
Ruszczak Z et al., 2003 [48]	Review	Primary literature on collagen as a biomaterial in drug delivery systems for antibiotics	Treatment and prophylaxis of bone and soft tissue infections	Incidence of SSI	The incidence of SSI was 4.3% in the treatment group and 9.0% in the control group (relative risk 0.47; 95% confidence interval 0.33–0.68; *p* < 0.001).
Friberg O et al., 2005 [32]	RCT	2000 cardio-vascular surgery patients	Standard prophylaxis combined with CCGI v/s standard alone (control)	Incidence of SSI	The incidence of SSI was 4.3% in the treatment group and 9.0% in the control group (relative risk 0.47; 95% confidence interval 0.33–0.68; *p* < 0.001).
Eklund AM et al., 2007 [33]	RCT	557 patients who underwent elective cardio-vascular surgery	Standard prophylaxis combined with CCGI v/s standard alone (control)	Incidence of SSI	Postoperative SSI occurred in 11 of 272 patients (4.0%) in the study group and in 16 of 270 patients (5.9%) in the control group. This difference was not statistically significant (*p* = 0.20).
Raja SG et al., 2012 [34]	RCT	9 publications on prophylactic use of GCCI in cardiovascular surgery	Adjunctive use of GCCI in prophylaxis of SSI	Morbidity associated with SSI following surgery	The adjunctive use of GCCI is particularly beneficial in high-risk patients and also cost saving.
Chang WK et al., 2013 [35]	Meta-analysis	6979 patients from major medical databases and trial registers for RCTs	Use of GCCI in prophylaxis of SSI	Endpoint of interest was the incidence of SSI	GCCI reduced SSI [OR = 0.51; 95% CI: 0.33–0.77; *p* = 0.001; number needed to treat (NNT) = 21; I = 75%]. These results were seen in subset analysis of clean-contaminated surgery (OR = 0.43; 95% CI: 0.20–0.93; *p* = 0.03; NNT = 9) specifically.
Modarai B et al., 2005 [49]	Single institution prospective randomized study	Fifty-nine upper limb PTFE grafts in 48 patients	Use of GCCI in prophylaxis of SSI	Incidence of SSI	The use of prosthetic material is associated with a poor overall patency rate and high risk of infective complications.
Costa Almeida CEP et al., 2014 [36]	Controlled Clinical Trial	60 patients with lower limb ischaemia	GCCI in the groin incision adjacent to the prosthesis	SSI rate and in-hospital days	GCCI use decreasing SSI rate and in-hospital days, and also reduce health care costs.
Rasheed et al., 2021 [50]	Review	Literature review of preventive strategies for groin SSI	Antimicrobial therapy and CCGI	Post-operative SSI	Collagen gentamicin implants are useful in preventing surgical site infection in the groin after vascular surgical procedures.
Holdsworth J et al., 1999 [37]	Single institution prospective randomized clinical trial	25 patients with infective and potentially infective complications of vascular bypass grafting	Use of GCCI in prevention and treatment of SSI	In situ prevention and treatment of graft infection	7/11 in situ treatments of a graft infection were successfully aborted; in the other 4 grafts were removed. None of the other 14 patients treated with GCCI subsequently had infective sequelae.

## Data Availability

Not applicable.

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
