# Peer review of "Groin Surgical Site Infection in Vascular Surgery: Systemic Review on Peri-Operative Antibiotic Prophylaxis"

_antibiotics, 2022, doi:10.3390/antibiotics11020134_

Round 1
Reviewer 1 Report
The authors describe a systematic review of antibiotic management in vascular surgeries involving the groin. There is value in this type of review, however, there are several major concerns that would need to be addressed before suitable for publication. I commend the authors for having this up to date as of December 2020. Thank you, please see below.
General:
There likely needs to be a better focus in the paper. It sometimes alludes to antibiotic management of SSIs, when really the focus is primarily on prophylactic or peri-operative antibiotics at the time of surgery. I would ensure that this is consistent throughout and is easily known by the readers. I would start with the title for example - this should state perioperative or prophylactic antibiotics. If this was not the intent of the authors to limit it, I do think it should be focused to prophylactic/peri-operative antibiotics, otherwise this is far too comprehensive of a review.
Another general comment is the organization of the sections. I have made some recommendations below regarding this.
Lastly, I believe more detail is needed along with interpretation from your expert authorship for many of the studies discussed.
Title:
I believe the words prophylactic or peri-operative should be in there describing antibiotics
Abstract:
There is really no evidence synthesized in the results, would recommend a few summary statements.
The conclusion is not truly just from this systematic review as this has been known for a while. I would instead conclude from your review specifically.
Introduction:
Need more discussion on epidemiology - it's not just Staph spp - gram negatives are a significant contributor. What about anaerobes? Fungal? All of this should be mentioned here for completeness. (may fit best at beginning of Page 2 of intro)
Consider a figure showing the epidemiology expected from these types of surgeries.
Line 39/40: The >15% at the end of that statement reads funny and >15% is extremely wide range as it implies 15.1-99%. Please rephrase and be more specific.
Page 2: Lines 49-54: This paragraph should move up before discussion of prophylactic antibiotics.
Page 2: Lin 55-56: Bacteria are not a risk factor. We are all colonized with Staph spp on our skin. I think this needs to be rephrased. They are especially not a modifiable risk factor.
Page 2: Lines 64/65: Bacteria do not increase patient M&M, bacterial infections perhaps?
Any comment on MSSA increasing and MRSA going down? This is certainly happening in the last 5 years.
Methods:
Please add observational studies and quickly define to your list of inclusion.
You may want to include a PRISMA chart for future peer review, although this is a suggestion only.
Results:
I would recommend 2 major things here:
Reorganization - I think there should be subsections describing the major points you researched through your systematic review. For example, "antibiotic selection", "antibiotic timing", "duration of prophylactic/peri-operative antibiotics", "Beta-lactam allergic patient"..etc. These will really help the reader. IF you decide to do it by surgical details (e.g. prosthesis present vs not), it will likely require you to add a table or more detail in the introduction about the different patient types/surgery.
Details - More details are needed for most studies discussed. The reader needs to know dosing, frequency, etc. I think breaking down by subsections may naturally allow for more details to be presented. These should include numbers and p-values where available.
Pre-op antibiotic therapy - this to me is the focus of the entire paper, or should be, so it shouldn't have it's own subsection as it should be implied. As mentioned before in the intro to increase discussion on epidemiology, there is some here that is applicable.
Page 5: LInes 145-148 seems out of place talking about gram-negative coverage. Again, reorganizing this by subsection/topical areas will help all of this.
Please discuss proper allergy reconciliation to eliminate (or drastically reduce) the need for vanc/clinda and use cefazolin. Cefazolin use is well described and the use of allergy reconciliation and/or with penicillin skin testing is well described. (Kufel et al. https://www.mdpi.com/2226-4787/7/3/136)
What about weight-based dosing or weight of patient influence on prophylactic antibiotics?
The details on page 7, in lines 211-218 are good. Further would be including drug selection/dosing in these studies. But I would include more detail for ref 27 for example.
The intraoperative gent is a standalone subsection and that is fine. Perhaps it could be more generally titled as "Intraoperative local antibiotic use". I also don't think you need the opening lines of each section (e.g. Lines 238-240) are repetitive.
Gentamicin should be spelled with an "i" not y throughout
Conclusions:
Prior to conclusions, I think the author interpretation and true synthesis of the data to make recommendations would be valuable. This could include gaps in the literature/needs for research and things are that are well concluded. This is done somewhat in the conclusions, but I think a separate section prior to conclusions would be valuable. Alternatively, this could be in each subsection.
What about national or international guidelines on the topic? How do these synthesize with your systematic review? That would be valuable to include or discuss if really not available/readily used.
Mention of stewardship? This is an important synergy between ID and vascular surgery. I would include this point.
Tables -
I like the tables, and they add value, but I would break them down by subsection or add columns that match your subsections you include in the results.
Author Response
Dear Reviewers,
We thank Reviewer n.1 for the important suggestions provided, which led to a major review carried out with the following specific changes:
- The title has been changed to define the major issues of the review;
- Epidemiology has expanded;
- The responsibility of gram-negative and fungal infections has been reported;
- The incidence reported in line 39-40 has been specified
- The position of the paragraph line 49-54 has been changed
- Line 55-56: the sentence has been better phrased
- Page 2 line 64/65: MMSA comment has been added and observational studies have been added to the inclusion list
- The review sections were reorganized, eliminating the post-operative antibiotic therapy section to focus on the title topic.
- The title of the section has been changed ("Intraoperative use of local antibiotics"), according to the instructions of the reviewer.
- We mentioned the use of skin tests to check allergies to cephalosporins and reduce the use of vancomycin in patients with suspected allergies (ref. Kufel)
- We have corrected the word "gentamicin" in the text and in the table.
Sincerely,
The Authors and the corresponding author Prof. Bruno Amato.
Reviewer 2 Report
The study was methodologically well-designed, nevertheless the aim, the results and especially the conclusion are to be revised.
In the abstract section:
The terms “appropriate and careful timely” need to be better detailed.
In my opinion the whole abstract should be remodulate, too many methodological details and general non-exhaustive conclusions.
It’s important to remark if post-antibiotic therapy is or not recommended:
- what type of antibiotic
- what the timing of re-dose,
- what screening strategies are adequate
Please remodulate also that sentence in the conclusion section: “We performed a 277 evidence-based literature review to characterize risk factors and determine solutions to 278 manage/eliminate inguinal SSI rates”.
What are the main risk factors?
Manage inguinal SSI? The management of SSI is not the aim of that study
Perhaps meant to prevent SSI rates?
Author Response
Dear Reviewers,
We thank Reviewer n.2 for the important suggestions provided, which led to a major review carried out with the following specific changes:
- We have defined the terms "appropriate and careful timely";
- We remodeled the abstract by reducing the methodological details and adjusting the conclusions; 3. The expediency of additional antibiotic therapy was defined;
- The conclusions have been reformulated according to the indications.
We hope that the corrections we made will now make the manuscript suitable for publication on Antibiotics, and we await confirmation from the Editor.
Sincerely,
The Authors and the corresponding author Prof. Bruno Amato.

Round 2
Reviewer 1 Report
The authors submit a revised version of the paper that addresses a number of the peer reviewer recommendations. The most major of which was reframing the paper to involve peri-operative antibiotics. The title and methods were updated, as well as the presentation of results in tables and introductory paragraphs. While I would still like to see the discussion portion organized a bit more with subtitles, that is not a must in my opinion.
Thank you to the authors for this revision.
Minor revision to abstract: I do think the last line in the objective portion of the abstract is out of place. I almost would just delete it.